# A Novel Core Effector Vp1 Promotes Fungal Colonization and Virulence of *Ustilago maydis*

**DOI:** 10.3390/jof7080589

**Published:** 2021-07-23

**Authors:** Cuong V. Hoang, Chibbhi K. Bhaskar, Lay-Sun Ma

**Affiliations:** 1Institute of Plant and Microbial Biology, Academia Sinica, Taipei 11529, Taiwan; hvcuong88@gmail.com (C.V.H.); chibbhibhaskar@gmail.com (C.K.B.); 2Molecular and Biological Agricultural Sciences Program, Taiwan International Graduate Program, National Chung Hsing University and Academia Sinica, Taipei 11529, Taiwan; 3Graduate Institute of Biotechnology, National Chung-Hsing University, Taichung 402, Taiwan; 4Biotechnology Center, National Chung Hsing University, Taichung 402, Taiwan

**Keywords:** *Ustilago maydis*, virulence, biotrophic pathogen, smut fungi, nuclear localization, core effector

## Abstract

The biotrophic fungus *Ustilago maydis* secretes a plethora of uncharacterized effector proteins and causes smut disease in maize. Among the effector genes that are up-regulated during the biotrophic growth in maize, we identified *vp1* (*virulence promoting 1*), which has an expression that was up-regulated and maintained at a high level throughout the life cycle of the fungus. We characterized Vp1 by applying in silico analysis, reverse genetics, phenotypic assessment, microscopy, and protein localization and provided a fundamental understanding of the Vp1 protein in *U. maydis*. The reduction in fungal virulence and colonization in the *vp1* mutant suggests the virulence-promoting function of Vp1. The deletion studies on the NLS (nuclear localization signal) sequence and the protein localization study revealed that the C-terminus of Vp1 is processed after secretion in plant apoplast and could localize to the plant nucleus. The *Ustilago hordei* ortholog UhVp1 lacks NLS localized in the plant cytoplasm, suggesting that the orthologs might have a distinct subcellular localization. Further complementation studies of the Vp1 orthologs in related smut fungi revealed that none of them could complement the virulence function of *U. maydis* Vp1, suggesting that UmVp1 could acquire a specialized function via sequence divergence.

## 1. Introduction

Smut fungi rely heavily on delivering diverse effector proteins to suppress plant immunity and modulate plant metabolic pathways to facilitate their infection [1,2,3]. In the genome of smut fungi, which infects most monocot grass species, approximately 7% of the coding genes encode proteins containing a signal peptide for secretion [4]. This number does not include the effectors that lack signal peptides and are predicted to be secreted via an unconventional secretion pathway [5,6]. Among the putative effector proteins, half of them have no functional domains [4]. This makes them novel and interesting candidates for studying their functional roles in fungal virulence.

Genome comparative analysis in the related smut fungi reveals that the effector genes are not well-conserved compared to other genes [7]. While some core effectors are present in smut fungi like *Ustilago maydis*, *Sporisorium scitamineum*, *Sporisorium reilianum*, *Ustilago*
*hordei*, and *Melanopsichium pennsylvanicum*, accessory effectors exist in a subset of smut fungi [4,8]. Several core effectors reported either having conserved or diverse functions from their orthologs. The previous studies on the *U. maydis* Pep1, Sta1, Cce1, and Rsp3 showed that the orthologs from related smut fungi could successfully rescue the reduced phenotype of the deletion mutants in *U. maydis* [9,10,11,12], while the anthocyanin-induction function of Tin2, or the virulence-promoting function of See1 and ApB73, could not be complemented by their orthologs [13,14,15]. This implies that the core effectors are under selection pressure to evolve rapidly, for different infection strategy purposes, or adapt to their hosts. The notion is further supported by the findings that Tin2 orthologs from *S. reilianum* and *U. maydis* evolve to target different kinase paralogs in maize [13].

The corn smut *U. maydis* induces anthocyanin and forms tumors in which the fungus proliferates and produces spores [16]. It secretes a large number of effector proteins that contribute to all the stages of the fungal pathogenic development in maize [3,8]. Most of the effectors are functionally uncharacterized and delivered into plants in continuous waves during biotrophic growth [1]. To date, only a few effectors have been functionally characterized. The *U. maydis* Pit2, Pep1, Rsp3, and Fly1 localize in maize apoplast, while Cmu1, Tin2, See1, Jsi1, and Mer1 translocate into plant cells. The apoplastic effector Pit2 inhibits maize cysteine proteases [17,18]. Pep1 interacts with maize peroxidase POX12 to suppress H_2_O_2_ production [19]. Rsp3 shields hyphae from the attack of at least two secreted maize antifungal proteins (AFP1 and AFP2) [12]. Fly1 is another apoplastic virulence factor that cleaves and deactivates the maize chitinases [20]. Cmu1 channels the salicylic acid precursor chorismate into the phenylpropanoid pathway and suppresses the SA biosynthesis [21]. Tin2 interacts with maize ZmTTK1 to induce anthocyanin production by concomitantly reducing plant lignification [22]. The interaction of the nuclear-localized effector See1 with the maize SGT1 protein reactivates DNA synthesis and cell division in infected leaves [23]. Jsi1 alters the jasmonate/ethylene signaling pathway by targeting the maize TOPLESS corepressor in the plant nucleus [24]. Mer1 promotes the auto-ubiquitination activity of plant E3 ligase RFI2 to inhibit host defense [25]. These studies revealed that *U. maydis* uses different strategies to suppress the plant’s defense system or manipulate the plant’s metabolic pathways. Despite the importance of effectors in *U. maydis* virulence, the majority of the effector proteins have not been functionally characterized. This hampers our understanding of the disease mechanisms of *U. maydis*. 

Here, we provide a fundamental understanding of a novel secreted effector protein, Vp1 (Virulence promoting 1), from *U. maydis*. The *vp1* gene is highly up-regulated during the biotrophic development of *U. maydis.* The C-terminal region containing the NLS of Vp1 is processed after secretion in the apoplast. We also provide evidence that the NLS is necessary to localize the Vp1 proteins to the plant nucleus and required for Vp1-promoting virulence function. Finally, we show that UmVp1 orthologs might be functionally diverse.

## 2. Materials and Methods

### 2.1. Strains, Growth Conditions, and Virulence Assays

*Zea mays* Honey 236 (Taiwan cultivar) was used for *U. maydis* infection, unless otherwise stated. The haploid solopathogenic strains SG200 [26] and all *U. maydis* strains used in this study are listed in Appendix A. *U. maydis* strains were grown on solid potato dextrose agar (PDA) (2.4% PD broth (Difco, Detroit, MI, USA) and 2% Bacto agar (Difco, Detroit, MI, USA)) plates. For plant infection, strains were grown in YEPSL liquid medium (0.4% yeast extract, 0.4% peptone, and 2% sucrose) to an OD_600_ of 0.8 at 28 °C. Cells were adjusted to a final OD_600_ of 1.0 in H_2_O before injected into stems of 7-day-old maize seedlings. The symptoms were scored at 10 days post infection (dpi). Disease symptoms of three independent infection assays were evaluated according to the previous reported disease rating criteria [16]. Disease index was the sum of values in each disease category, calculated by multiplying the number of plants in one category with the value assigned to the category (Death plant =  11; Heavy tumor on base of stem or stunted growth  =  9; Normal tumor on leave and/or steam  =  7; Small tumors  =  5; Chlorosis  =  3; No symptoms  =  1) and divided by the total infected plants per strain [18]. Significant differences of disease symptoms in each strain were compared with SG200 or *vp1* deletion mutant, indicated in each figure legend using a two-tailed Student’s *t*-test. 

### 2.2. Plasmid and Strain Construction 

All of the primers and plasmids, generated using either Gibson Assembly or standard cloning methods, in this study are listed in Appendix A. The manufacturer’s protocol from New England Biolabs was used for Gibson Assembly. For standard cloning, the PCR fragments were digested by restriction enzymes and ligated into a vector using T4 DNA ligase (NEB Biolabs, Evry, France).

To generate the *vp1* mutant, the 1 kb fragments, from upstream and downstream of *vp1* genes, were amplified from SG200 genomic DNA using primer pairs #1/2 and #3/4. The two PCR fragments were combined with a *SfiI*-digested hygromycin resistance cassette, isolated from pBS-hhn [27], and inserted into the *EcoR*V-digested pJet vector using Gibson Assembly to yield the pJet-*vp1*KO plasmid. The *Ssp*I-digested pJet-*vp1*KO plasmid yielded a DNA fragment containing 5′UTR-hygromycin cassette-3′UTR and was transformed into SG200 to create SG200Δvp1.

For the complementation of the Δ*vp1* mutants, the DNA was cloned into a p123 vector [28] and linearized by *Ssp*I before transformed into Δvp1 protoplasts. Transformants containing a single copy of the gene integrated into the *ip* locus [29] were screened by Southern blot analysis. 

Plasmid pvp1 was generated by inserting a PCR fragment (primer pair #5/6) containing about 1 kb native *vp1* promoter region and the entire open reading frame (ORF) into the *Nde*I/*Not*I-digested p123. Plasmid pHA-vp1 was created by ligating *Nde*I/*Not*I-digested p123 with the two PCR fragments, which amplified from pvp1 using the two primer pairs (#7/8 and #9/10). A PCR fragment amplified from pHA-vp1 using the primer pair #11/6, was digested and inserted into *Nco*I/*Not*I-digested p123 to generate potef-HA-vp1. To generate plasmid gfp-vp1, three PCR fragments were amplified using primer pairs #7/12, #13/14, and #15/16 and inserted into *Nde*I/*Not*I-digested p123 using Gibson Assembly. Plasmid vp1-gfp was generated via the same strategy, using primer pairs #7/17 and #18/19. To generate the ortholog complementation strains, the *Nde*I/*Not*I-digested p123 was ligated with the vp1 promoter fragment amplified using #7/20 and a fragment containing the entire open reading frame (ORF) of the vp1 orthologs from *U. hordei*, *S. reilianum*, *S. scitamineum*, and *M. pennsylvanicum* amplified using the primer pairs #21/22, #23/24, #25/26, and #27/28, respectively. To create pSrvp1 and pUhvp1 using a signal peptide of UmVp1, the *Nde*I/*Not*I-digested p123 was ligated with the two fragments amplified using the primer pairs #7/29 and #30/24 for pSrvp1 or the two fragments amplified using the primer pairs #7/31 and #32/22 for pUhvp1. To generate the plasmid pvp1-nls*, the two DNA fragments containing point mutations were amplified using two pairs of primers #7/33 and #34/35 from the pvp1 plasmid and followed by Gibson Assembly. To generate plasmid p vp1(Δnls), the vp1(Δnls) fragment was amplified by primer pair #5/46, digested by *Nde*I*/Not*I, and ligated into *Nde*I/*Not*I-digested p123. Additionally, potef-dSP-HA-vp1 was created by ligating the #45/6-amplified fragment into *Xma*I/*Not*I-digested p123 and potef-vp1-HA was created by Gibson Assembly using #11/47 and #48/35-amplified fragments and *Xma*I/*Not*I-digested p123; this plasmid was then cut by *Xma*I/*Xba*I and ligated with the #11/49-amplified fragment to create potef-vp1(Δnls)-HA. Additionally, mCherryHA and mCherry were amplified by pairs #50/51 and #50/53 (respectively) and then digested by *Xba*I/*Not*I and ligated into *Xba*I/*Not*I-digested potef-vp1-HA to yield potef-vp1-mCherryHA and potef-vp1-mCherry. To create potef-vp1(nls*)-mCherryHA, the *Xma*I/*Xba*I-digested fragment amplified from pvp1(nls*) using primer pair #11/52 and was ligated into *Xma*I/*Xba*I-digested potef-vp1-mCherryHA.

To express GFP-tagged Vp1 in maize protoplasts and tobacco leaves, the binary vector pEZRK was *Sal*I/*Xba*I-digested and ligated with the primer pair #36/37-amplified fragment to generate pgfp-vp1 or ligated with the primer pair #38/39-amplified fragment to generate pvp1-gfp, or ligated with the primer pair #36/42-amplified fragment to generate pvp1-gfp(nls*). To generate pgfp-Uhvp1, the *XbaI* digested gfp-vp1 plasmid was ligated with a PCR fragment amplified from pUhvp1 using the #40/41 primer pair. The rfp-virD2nls plasmid was generated by ligating the *Sal*I/*Xba*I-digested pEZRK and the PCR product amplified using primers #43/44.

### 2.3. Quantitative Real-Time PCR

To determine the gene expression of *vp1*, quantitative real-time RT-PCR was performed, as described [12]. The SG200-infected leaves were ground to a fine powder in liquid nitrogen, and the total RNA was extracted using a TRIzol reagent (Life Technologies). The genomic DNA was removed from the total RNA using a TURBO DNA-free™ Kit (Ambion). The cDNA was reverse-transcribed using a SuperScript^®^ III First-Strand Synthesis SuperMix (Invitrogen). All of the cDNA samples were diluted to 20 ng/ul before qRT-qPCR analysis. The primer pairs used in qRT-PCR are listed in Appendix A. The expression of the constitutively expressed *U. maydis peptidyl-prolyl isomerase* (*ppi*) gene was used to normalize the expression of the gene indicated. The relative expression values of *vp1* were calculated using the 2^−^^ΔΔCt^ method [30]. The fungal biomass analysis was performed using genome DNA, as described previously, with some modifications [31]. A 2-cm long-segment, below the injection holes of SG200-or SG200Δvp1-infected maize leaves, was used to prepare the genomic DNA for qPCR analysis. The expressions of the *U. maydis*
*ppi* and *Z. mays*
*GAPDH* genes were used to calculate the relative fungal biomass. Three biological replicates were performed. The primers for *GAPDH* are listed in Appendix A. The statistical analysis between the two different samples was determined by a two-tailed Student’s *t*-test.

### 2.4. Secretion Assay and Apoplastic Fluid Protein Precipitation

The secretion assay of *U. maydis* was performed as described previously [21]. The *U. maydis* cells were grown in YEPSL medium to an OD_600_ of 0.6. The cell pellets were collected and suspended in a sample buffer (50 mM Tris-HCl, 2% SDS, 10% glycerol, 100 mM DTT, and 0.01% bromophenol blue). The culture supernatants were trichloroacetic acid-precipitated, acetone-washed, and, finally, suspended in a sample buffer. The total proteins from the cell pellets and supernatant fractions were separated by SDS-PAGE and detected by Western blot analyses using mouse antibodies against HA or tubulin epitopes. To collect apoplastic fluids from infected leaves, a vacuum-centrifugation method was applied and followed by TCA-precipitation, as described previously [12].

### 2.5. Subcellular Localization

To determine the subcellular localization of Vp1, maize protoplasts were isolated and transformed, as described in rice [32], with some modifications. The 0.5 mm maize leaf fragments were incubated in the enzyme solution (10 mM MES (pH 5.7), 0.6 M mannitol, 1.5% Cellulase RS, 0.75% Macerozyme, 0.1% BSA, and 1 mM CaCl_2_] for 4 h in darkness, with gentle shaking. The protoplasts were collected and washed with W5 solution (2 mM MES pH 5.7, 154 mM NaCl, 125 mM CaCl_2_, and 5 mM KCl) and re-suspended in MMG solution (4 mM MES pH 5.7, 0.6 M mannitol, and 15 mM MgCl_2_). The nuclear marker plasmid rfp-virD2nls and the no-signal, peptide-containing plasmids gfp-vp1, vp1-gfp, gfp-Uhvp1, or gfp-vp1-nls* were co-transfected into maize protoplasts using a 40% PEG solution (0.6 M mannitol, 100 mM CaCl_2_, and 40% *v*/*v* PEG). The protoplasts were washed and re-suspended in W5 solution and incubated at 28°C in darkness overnight. The fluorescent signals of the fusion proteins were detected using confocal microscopy.

### 2.6. Microscopy

The infected leaf tissue was de-stained in 100% ethanol and incubated in 10% KOH at 85 °C for 4 h. The leaves were washed with 1× PBS (pH7.4) and stained with propidium iodide (Sigma Aldrich, Steinheim, Germany) and WGA-AF488 (Molecular Probes, Eugene, OR, USA) to visualize the plant cell and fungal hyphae, respectively. The WGA-488 fluorescent was excited at 488 nm and detected at 500–540 nm. The propidium iodide was excited at 561 nm and detected at 580–630 nm. The Vp1 fusion protein localization in the maize protoplasts was also observed by confocal microscopy (Leica Microsystems, Wetzlar, Germany). The GFP signal was visualized with excitation at 488 nm and emission at 500–550 nm; the RFP signal was visualized with excitation at 561 nm and emission at 570–630 nm.

### 2.7. Bioinformatic Analysis

The gene and protein sequence of UMAG00538 (UmVp1) can be accessed at the fungal genome resourse (https://mycocosm.jgi.doe.gov/mycocosm/home, accessed on 15 March 2021). For the ortholog identification using reciprocal-best BLAST hits, the accession numbers of the Vp1 orthologs identified from blastP analysis using the UmVp1 sequence were used in non-redundant BLASTP searches against the *U. maydis* 521 (taxid: 237631) sequence database. Accession number of orthologs: *XP_011386375.1* (*U. maydis*) [16], *XP_012192856.1* (*Pseudozyma hubeiensis*) [33], *CBQ67951.1* (*Sporisorium reilianum*) [7], *CDU24160.1* (*Sporisorium scitamineum*) [34], *SAM61624.1* (*Ustilago bromivora*) [35], *SPO21134.1* (*Ustilago trichophora*) [36], *CAJ41947.1* (*Ustilago hordei*) [37], *CDI51462.1* (*Melanopsichium pennsylvanicum*) [38], *ETS63783.1* (*Moesziomyces aphidis*) [39], *XP_014659418.1* (*Moesziomyces antarcticus*) [40], and *PWY98796.1* (*Testicularia cyperi*) [41]. The secretion signal peptide was predicted by SignalP 5.0 (http://www.cbs.dtu.dk/services/SignalP/, accessed on 15 March 2021) [42]. Vp1 Orthologs retrieved from OrthoDB search (https://www.orthodb.org/?ncbi=1&query=23561808, accessed on 1 July 2021) [43]. Nuclear localization signal sequences were predicted by LOCALIZER (http://localizer.csiro.au/, accessed on 15 March 2021) [44]. Sequence alignments were generated using Clustal Omega (https://www.ebi.ac.uk/Tools/msa/clustalo/, accessed on 15 March 2021) [45].

## 3. Results

### 3.1. U. maydis UmVp1 Is an Important Virulence Factor

Based on the time-resolved RNAseq data generated using maize leaves infected by two compatible strains, FB1 and FB2, approximately 445 predicted effector genes, including *umag00538*, were differentially expressed only during plant-associated stages [1] (Appendix A). This may indicate that UMAG00538 is one of the virulence factors required for fungal development inside plant cells. We designate this effector protein Vp1 (Virulence promoting 1). Using in silico analysis, *vp1* is a single gene located on chromosome 1 the of *U. maydis* genome and encodes a protein consisting of 354 amino acids (Figure 1A). The protein features a signal peptide, predicted by SignalP 5.0 [42], and a putative nuclear localization signal (NLS) sequence embedded at the C-terminus, identified by LOCALIZER [44]. However, Vp1 lacks any known functional or structural domains, which provides no hints to reveal the molecular function.

To characterize Vp1 in the parental solopathogenic haploid strain SG200 that does not require a mating partner [16], we performed a qRT-PCR analysis of the *vp1* gene expression in SG200-infected leaves. The *vp1* gene was not expressed in axenic culture, was significantly up-regulated at the early stage, and maintained high expression throughout the biotrophic development of SG200 (Figure 1B). The expression patterns of the *vp1* gene in both SG200 and FB1xFB2 infected leaves were comparable (Figure 1B and Appendix A). We further analyzed the expression of *vp1* in SG200 infected tassels to examine if the *vp1* gene is organ-specific-induced. While the *U. maydis see1* is specifically expressed in infected maize leaves, but not in tassels [23], the *vp1* gene was highly expressed in both infected leaves and tassels (Figure 1B and Appendix A). The results suggest that the expression of the *vp1* gene is induced during the biotrophic phase and has no organ-specificity.

We next examined the contribution of Vp1 to the *U. maydis* virulence. The disease symptoms of SG200, the *vp1* deletion mutant, and the complementation strains developed on the maize varieties Honey 236 (Taiwan cultivar) and Early Golden Bantam were accessed. The deletion of the *vp1* gene resulted in a significant reduction of virulence compared to SG200 (Figure 1C and Appendix A). The reduced virulence phenotypes of the *vp1* mutant could be fully rescued by complementation with a single copy of the *vp1* allele, carrying either no tag or HA-tag, after the downstream of the N-terminal signal peptide (Figure 1C), suggesting the importance of Vp1 in promoting the disease.

To investigate whether the deletion of *vp1* affects fungal colonization in plant leaves, we collected infected leaves at 3 and 6 dpi and performed fungal biomass assays. The quantitative PCR analysis on total genomic DNA showed that the fungal biomass of SG200Δvp1 strain at 6 dpi reduced significantly compared to SG200. However, there was no significant difference between the two strains at 3 dpi (Figure 1D). We further investigated the plant colonization of the SG200Δvp1 and SG200 strains via WGA-AF488/PI-staining using confocal microscopy. The SG200Δvp1 strain showed a noticeable reduction in the fungal colonization in infected leaves from 6 dpi onwards. However, there was no significant difference before 4 dpi (Figure 1E and Appendix A). Both fungal biomass analysis and microscopic data analysis demonstrated that Vp1 is necessary for fungal colonization.

### 3.2. UmVp1 Is Secreted by U. maydis

The *U. maydis* UmVp1 is predicted to consist of a 29 amino acids signal peptides (SP) at its N-terminus (Figure 1A), suggesting the secretion of Vp1 by *U. maydis*. To visualize the secretion in vitro, we generated the two complementation strains SG200Δvp1 that constitutively expressed Vp1, tagged with a HA-epitope downstream of SP (HA-Vp1), and a N-terminal HA-tagged Vp1, without a signal peptide (dSP-HA-Vp1) under an *otef* promoter. While no protein band in the SG200 sample was detected in the anti-HA immunoblots, the HA-Vp1 and dSP-HA-Vp1 protein bands could be detected in the cell pellet fractions. However, in the supernatant fractions, the HA-Vp1, but not dSP-HA-Vp1, protein bands were detected (Figure 2A). The detection of the different HA-Vp1 protein species in the supernatant fraction suggests possible processing at the C-terminus of the Vp1 proteins after secretion (or the cause of protein stability). The result indicates that the signal peptide of Vp1 is functional and required for protein secretion. We also observed that the HA-Vp1 proteins migrated slower than expected. It has an apparent molecular weight of 55 kDa, which is higher than the expected size (38 kDa). Since the Vp1 proteins do not show N- and O-glycosylation (Appendix A) or disordered proteins, including the *U. maydis* effector Rsp3, which has been reported to migrate anomalously [12,46,47,48], we thereby analyzed the secondary structure of Vp1 using the online bioinformatics tools. The D^2^P^2^ database (http://d2p2.pro/search, accessed on 15 March 2021) and Phyre^2^ server (http://www.sbg.bio.ic.ac.uk/~phyre2/, accessed on 15 March 2021) predict a disordered, unstructured region at the C-terminus of Vp1 (amino acids 210–354; 57% disordered; Appendix A). The anomalous migration pattern of Vp1 is more likely caused by the predicted disordered region or some unknown post-translational modification.

To visualize the secretion of Vp1 *in planta*, we generated the complementation strains expressing GFP-Vp1 and Vp1-GFP fusion proteins and infected maize seedlings. While expressing Vp1-GFP proteins could fully rescue the reduced virulence phenotype of the *vp1* mutant, the virulence of the GFP-Vp1 complementation strain did not reach wild-type levels (Figure 2B). The GFP signals of fusion proteins could be detected inside the fungal cytoplasm and was found to accumulate around the fungal hyphal tips (Figure 2C), where the secretory vesicle supply center, Spitzenkörper, is located [49,50,51]. The localization patterns of Vp1-GFP and GFP-Vp1 fusion proteins were distinct from the cytosolic GFP proteins that localize inside hyphae. The observations of the Vp1 proteins accumulating at the hyphal tips are in line with other reported *U. maydis* secreted effectors that showed similar accumulation patterns during biotrophic growth [52,53].

We also performed plasmolysis experiments to visualize the fluorescence of secreted Vp1-mCherry, mCherry-AvitagHA, and the non-secreted cytosolic mCherryHA proteins in enlarged apoplastic spaces of the infected maize leaves. While the diffusible fluorescence of the secreted mCherry-AvitagHA proteins could be visualized in the expanded spaces, the non-secreted, cytosolic mCherryHA fluorescence was restricted inside the fungal hyphae (Appendix A). Compared to the mCherry-AvitagHA fluorescence, the Vp1-mCherry protein fluorescence was weakly visualized in the expanded plasmolysis area (Appendix A). The immunoblotting analysis of the total cell lysate fractions of the infected leaves showed that the full-length mCherry-AvitagHA and Vp1-mCherry proteins were expressed (Appendix A). However, in the TCA-precipitated apoplastic protein fractions, the full-length Vp1-mCherry proteins were barely detected (Appendix A). Since the truncated mCherry fragments were detected in both sample lanes of the apoplastic fractions, the Vp1-mCherry fusion proteins might not be stable (or there is a C-terminal processing of the Vp1 proteins). Altogether, these results illustrate that Vp1 secretes from *U. maydis*.

### 3.3. NLS Is Important for Vp1 Function and the Nuclear Localization

A putative NLS is predicted at the C-terminus of *U. maydis* Vp1 (Figure 1A and Appendix A), implying that Vp1 may localize to the plant nucleus. We failed to visualize the localization of the GFP-tagged Vp1 proteins in the nucleus of maize leaves infected by the complementation strain-expressing, GFP-tagged Vp1 proteins. We, thus, expressed the GFP-tagged Vp1 without signal peptide, as well as the nuclear marker RFP-VirD2NLS proteins in maize protoplasts prepared from SG200Δvp1 infected leaves (Figure 3A) and healthy leaves (Appendix A) via polyethylene glycol (PEG)-mediated transformation. We detected that the fluorescence of GFP-Vp1 and Vp1-GFP was highly accumulated in the nuclei of the maize protoplasts and in some of the plant cytoplasm (Figure 3A and Appendix A). By substituting the amino acids lysine and arginine in the NLS sequence, that are the key amino acids for nuclear localization [54], with alanine (NLS*), the signals of GFP-Vp1(NLS*) proteins mainly detected plant cytoplasm and fewer signals found in maize nucleus (Figure 3A and Appendix A). Similar localization patterns were observed when the GFP fusion proteins Vp1 and Vp1 (NLS*) were transiently expressed in *Nicotiana benthamiana* leaf cells using the Agrobacterium infiltration method (Appendix A). These results show that the NLS of Vp1 protein is functional and could localize the Vp1 proteins to the plant nucleus if the Vp1 translocates to plant cells.

We next measured if Vp1(NLS*) could complement the virulence of SG200Δvp1 when the nuclear localization is almost blocked. Vp1(NLS*) could only partially complement the virulence phenotype of the *vp1* mutant (Figure 3B). As no statistically significant difference was found when comparing the SG200Δvp1-vp1(nls*) with SG200, as well as with SG200Δvp1, we considered that the Vp1(NLS*) complementation strain showed an intermediate phenotype. It could be due to the mutations in the NLS sequence, which do not completely abolish the nuclear localization of Vp1(NLS*) proteins. To further explore the importance of the NLS in Vp1 function, we generated the complementation strains expressing Vp1 proteins without the NLS (Vp1ΔNLS). The Vp1ΔNLS proteins could not complement the virulence phenotype of Δvp1 (Figure 3C), suggesting the importance of the NLS in the Vp1 virulence function.

### 3.4. Vp1 Is Processed at the C-Terminus after Secretion

To understand how NLS affects the Vp1 protein function and whether mutations or deletion of NLS have any impacts on the protein secretion, we visualized the secretion of the Vp1 proteins. Despite the fact that Vp1-HA and Vp1ΔNLS-HA proteins could be detected in the cell pellet fractions of the strains that constitutively expressed the indicated proteins, the full-length Vp1ΔNLS-HA (but not Vp1-HA proteins unexpectedly found in the supernatant fractions (Figure 3D). An immunoblot analysis of apoplastic proteins from infected leaves showed that Vp1-HA proteins were not detectable while the Vp1ΔNLS-HA and the positive control *U. maydis chorismate* mutase Cmu1 accumulated in the apoplastic protein fraction of infected maize leaves [12] (Figure 3E).

While the N-terminal HA-tagged Vp1 proteins (but not the C-terminal HA-tagged Vp1) were detected in the supernatant fractions, we speculate the possible processing at the C-terminus of the Vp1 proteins. We, thus, tagged a larger size protein, mCherryHA, at the C-terminus of Vp1 to visualize the C-terminal tagged Vp1 proteins. The full-length proteins of Vp1-mCherryHA and Vp1(NLS*)-mCherryHA were detected only in the cell pellet fractions, but 34 kDa bands corresponding to the possible C-terminal processed forms of Vp1 fusion proteins were found in the supernatant fractions (Figure 3F). These results suggest that the processing occurs at the C-terminal of Vp1 after secretion, and the deletion of the NLS affects the protein processing.

### 3.5. The Vp1 Orthologues in Smut Fungi Are Not Functionally Conserved

Based on a BLASTP analysis and reciprocal best BLAST hits against *U. maydis* genome, we identified the putative orthologs of Vp1 in related smut-fungi with a coverage of 60–94% and 45–60% identity (Appendix A). Among the eleven putative orthologs, six Vp1 orthologs from *U. maydis* [16], *U. hordei* [37], *Sporisorium reilianum* [7], *Sporisorium scitamineum* [34], *Moesziomyces antarcticus* [40], and *Pseudozyma hubeiensis* [33] could further be retrieved from the ortholog database search OrthoDB (https://www.orthodb.org/?ncbi=1&query=23561808, accessed on 1 July 2021) [43]. The amino acid sequences of the Vp1 orthologs display a relatively high degree of identity in the N-terminal sequences (55–71%), while the C-terminal regions are more diverse (33–50%) (Appendix A).

The putative Vp1 orthologs that lack NLS at their C-termini could imply their distinct subcellular localization (Appendix A). However, they might hijack plant nuclear proteins and localize to the plant nucleus, as has been reported in some nuclear-targeting effectors [55,56]. To examine if the Vp1 orthologs lacking NLS could localize to the plant nucleus, we selected *U. hordei* UhVp1 for further analysis. The signals of GFP-UhVp1 could only be visualized in the cytoplasm of the maize protoplasts and *N. benthamiana* leaf cells, and they were barely detected in the nuclei (Figure 3A, Appendix A). The localization pattern of the UhVp1 protein was distinct from the UmVp1 proteins, implying that they might not have a conserved function, which prompted us to investigate the possibility of the putative Vp1 orthologs in rescuing the SG200Δvp1virulence function.

The putative Vp1 orthologs, with or without NLS from the close-related smut fungi, were selected to complement the virulence of the SG200Δvp1 strain. The expression of the Vp1 orthologs in the SG200Δvp1 strains was under the control of the *umvp1* promoter, and the secretion was mediated by their signal peptide (Figure 4A) or by the UmVp1 signal peptide (Figure 4B). Surprisingly, none of them rescued the virulence phenotype of SG200Δvp1 back to the level of the wild-type SG200 (Figure 4), even though the *srvp1* gene expression could be detected in the infected tissues (Appendix A), suggesting that UmVp1 is functionally specialized in *U. maydis*.

## 4. Discussion

In this study, we demonstrate that Vp1, a novel secreted effector of *Ustilago maydis,* is important for the fungal colonization and virulence in maize. It has neither maize-line nor organ specificity. The C-terminal NLS embedded region of Vp1 could be processed after secretion in plant apoplasts. The NLS sequence is required for the virulence-promoting function of Vp1 and could localize the protein to the plant nucleus if the Vp1 translocates to plant cells.

The fungal biomass of the *vp1* mutant is not significantly affected at the early stages of biotrophy (before 4 dpi). It could be reasoned by the high expression levels of Vp1 at the later stages and is most likely involved in the late stage of biotrophy, the phenotype of the *vp1* mutant could, thus, be noticeable at later stages. However, it is plausible that the disruption of the *vp1* gene at the early stage does not show a discernable difference in fungal biomass, which could be due to the limitations and sensitivity of the phenotypic assays.

Due to the low expression level of the Vp1 proteins, we have not successfully demonstrated the translocation of Vp1 to the plant cells using microscopic analysis and subcellular fractionation assays. We showed the functionality of the NLS in guiding the Vp1 proteins to the plant nucleus, based on the assumption that the proteins could translocate to plant cells. We also demonstrated the importance of the NLS in the Vp1-promoting virulence function with the virulence assessment analysis on the vp1(Δnls) mutant. This could be caused by Vp1(ΔNLS) that could not be processed. The detection of the Vp1 proteins in plant apoplasts provides evidence that the Vp1 proteins reside in the apoplasts, in addition to the plant nucleus. The apoplast-localization could also be a transit point for translocated proteins heading to their final destination. This is evident by the translocated effectors Cmu1 and Tin2, which accumulate in apoplast before translocating inside plant cells to exhibit their function. However, the findings from processing Vp1 suggest that it could process at least into two regions, pointing to a possibility that the relevant virulence function of Vp1 could take place in either the nucleus or apoplasts (or both). Even though the processing sites of Vp1 have not yet been identified, the processing is important for Vp1 function. Blocking the processing might result in non-functional Vp1 proteins and restrict the proteins in apoplasts. The identification of the target of Vp1 will help to decipher the molecular mechanism of Vp1 proteins and fill these gaps in the future.

Mutations of the NLS sequence restrict most Vp1 proteins in the plant cytoplasm and partially complement the virulence phenotype of the vp1 mutant. We speculate that since the processing of Vp1(NLS*) proteins is not blocked, the other processed regions could still be functional and exhibit their action, even though the NLS is mutated. Additionally, mutations in the NLS could allow a low amount of proteins to act in the nucleus to restore the protein function.

Putative Vp1 orthologs are present in all Ustilaginaceae and in *Testicularia cyperi*, which belongs to Anthracoideaceae family. Only certain orthologs contain a predicted NLS sequence in the highly diverse disordered C-terminal regions, which show a relatively low sequence identity (33–50%). The cross-species complementation fails to restore the virulence function of the *U. maydis vp1* mutant, regardless of the presence of NLS. The high variability of the Vp1 orthologs in the related smut species, e.g., *S. reilianum* and *U. hordei*, suggests an evolutionary pressure on this effector and reflects the importance of Vp1 in biotrophic fungal virulence. The unsuccessful complementation, and the finding of the cytoplasmic localization of *U. hordei* ortholog UhVp1 lacking a predicted NLS, suggests that the Vp1 orthologs might interact with different targets located in the same, or different, cell compartments to promote fungal virulence. Unsuccessful cross-species complementation has also been reported for *U. maydis* See1, AbB73, and Tin2. In the case of the nuclear-localized, organ-specific effector UmSee1, the orthologs UhSee1 and SrSee1 fail to complement the virulence function of UmSee1 [14,57]. The orthologs UbApB73 and MpApB73, but not SrApB73, fail to restore the ApB73 mutant phenotype [15]. UmTin2 has been neofunctionalized and stabilized ZmTTK1 to induce maize anthocyanin when SrTin2 targets *ZmTTK2* and *ZmTTK3* [13,22]. The functional divergence of the effectors could arise from a common ancestor over evolutionary time under selection pressure to develop new invasion strategies in benefiting distinct pathogenic lifestyles [58]. These effector orthologs might likely interact with different family members to facilitate their needs in specific hosts. However, how the Vp1 orthologs are diverse in function is yet unclear. Intrinsically disordered regions are known to be more persistent than ordered regions when undergoing amino acid substitutions/insertions/deletions and the endowment of proteins with functional promiscuity [59,60,61]. We speculate that the C-terminal disordered region of Vp1 proteins is more likely to evolve to create functional diversification.

In summary, our work provides a fundamental understanding of one of the novel effectors in *U. maydis*. Vp1 is processed and could reside in plant apoplast and/or might translocate to the plant’s cytosol and nucleus to exhibit the virulence-promoting function. Identifying the target of Vp1 to decipher the molecular mechanism and resolve the mystery of the protein function will be the next challenge in the future.

## Figures and Tables

**Figure 1 jof-07-00589-f001:**
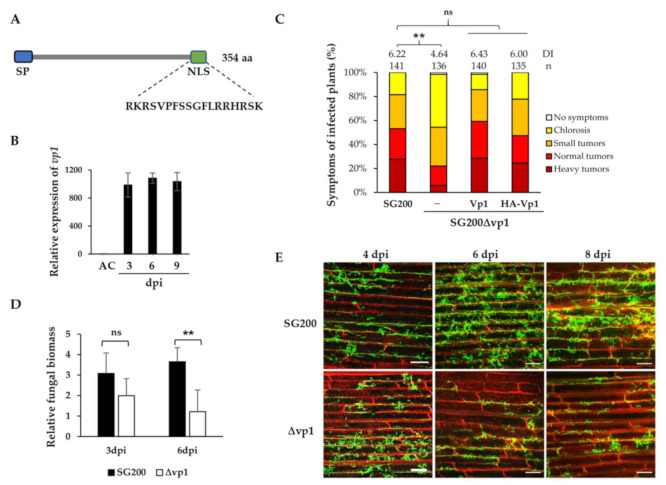
UmVp1 is an important virulence factor of *U. maydis. (***A**) Schematic drawing of Vp1 with a predicted N-terminal signal peptide (SP) and C-terminal nuclear localization signal (NLS). The predicted NLS sequence is shown. (**B**) Gene expression profiling of *vp1* by qRT-PCR. Total RNA was extracted from SG200 infected maize leaves which were collected at 3, 6, and 9 days post-infection (dpi) and from cells grown in axenic culture (AC). The constitutively expressed *U. maydis* peptidylprolyl isomerase (*ppi*) gene was used for normalization. The data represent the mean obtained from three independent experiments and the error bars depict ± SD. (**C**) The 7-day-old maize seedlings were infected with the indicated complementation strains and disease symptoms were scored at 10 dpi; n: indicates total numbers of infected plants from the three independent infection assays; DI is the average disease index value of three independent infection assays; ** *p* < 0.01 indicates significant differences of disease symptoms in respective complementation strains compared with SG200 determined by a two-tailed Student’s *t*-test using the disease index values. (**D**) *U. maydis ppi* gene and *Z. mays gapdh* gene were used to calculate relative fungal biomass of the indicated strains in the infected leaves at 3 and 6 dpi. The data represent the mean obtained from three independent infection assays and the error bars depict ± SD; ns indicates “not significant”, ** *p* < 0.01. (**E**) Maize leaves infected by strains SG200 and SG200Δvp1 were collected at the indicated dpi and were stained with WGA-AF488 (green) and propidium iodide (red) to visualize fungal hyphae and plant cell walls, respectively. Bars: 200 µm.

**Figure 2 jof-07-00589-f002:**
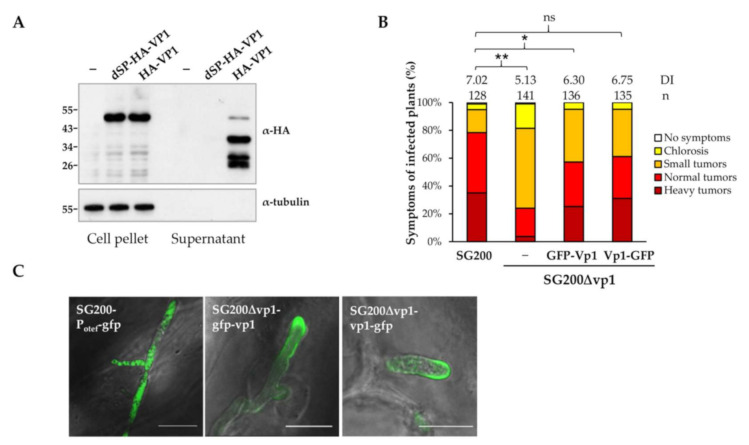
Vp1 is secreted by *U. maydis*. (**A**) SG200 (indicated as -) and SG200Δvp1 strains, expressing the indicated proteins under a constitutive promoter *otef*, were grown in YEPSL medium to an OD_600_ of 0.6; dSP: deletion of signal peptide. Proteins from cell pellets and supernatants (TCA-precipitation) were subjected to western blot analysis. The western blots were developed using the antibody against HA and tubulin epitopes. Cytosolic protein Tubulin served as a negative control for secretion. (**B**) The 7-day-old maize seedlings were infected with the indicated complementation strains, and the disease symptoms were scored at 10 dpi; n indicates total numbers of infected plants from the three independent infection assays. DI is the average disease index of three independent infection assays. ns, not significant. * *p* < 0.05 and ** *p* < 0.01 indicate significant differences of disease symptoms in respective complementation strains compared with SG200 determined by a two-tailed Student’s *t*-test using the disease index values. (**C**) Secretion of Vp1 *in planta*. The GFP fluorescence from leaves infected with the indicated multiple-copy gene integration strains expressing cytosolic GFP (no SP) under constitutive promoter *otef* and expressing GFP-Vp1 and Vp1-GFP under native promoter at 5 dpi were observed using confocal microscope. Bar, 10 µm.

**Figure 3 jof-07-00589-f003:**
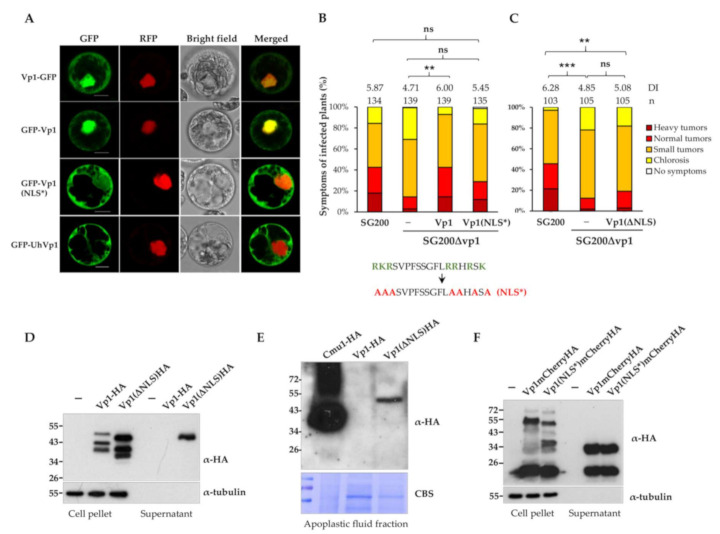
*U. maydis* Vp1 proteins localize to plant cytoplasm and nucleus. (**A**) Fluorescent signals of GFP-tagged Vp1 and the nuclear marker RFP-VirD2NLS co-expressed in maize protoplasts prepared from the Δvp1-infected leaves were visualized by confocal microscopy. Bars, 10 µm. (**B**,**C**) The maize seedlings that were infected with the indicated strains and disease symptoms were scored at 10 dpi; n indicates total numbers of infected plants from the three independent infection assays. DI is the average disease index value of three independent infection assays; ns, not significant. *** *p* < 0.001 and ** *p* < 0.01 indicate significant differences of disease symptoms in respective complementation strains, compared with either SG200 or SG200Δvp1, determined by a two-tailed Student’s *t*-test using the disease index values; ns, not significant. Amino acids indicated in green in the NLS sequence were substituted with alanine (red). The mutated sequence designated NLS*. ΔNLS: deletion of NLS. (**D**,**F**) SG200 (indicated as -) and SG200Δvp1 strains expressing the indicated proteins under a constitutive promoter *otef* were grown in YEPSL medium to an OD_600_ of 0.6. Proteins from cell pellets and supernatants (TCA-precipitation) were subjected to western blot analysis. The western blots were developed using the antibody either against HA and tubulin epitopes. Cytosolic protein Tubulin served as a negative control for secretion. (**E**) TCA-precipitated apoplastic fluids collected from maize leaves infected with SG200-P_otef_-cmu1HA, SG200Δvp1-P_otef_-vp1HA, and SG200Δvp1-P_otef_-vp1(Δnls)HA strains at 3 dpi; CBS, Coomassie Blue Staining as loading control.

**Figure 4 jof-07-00589-f004:**
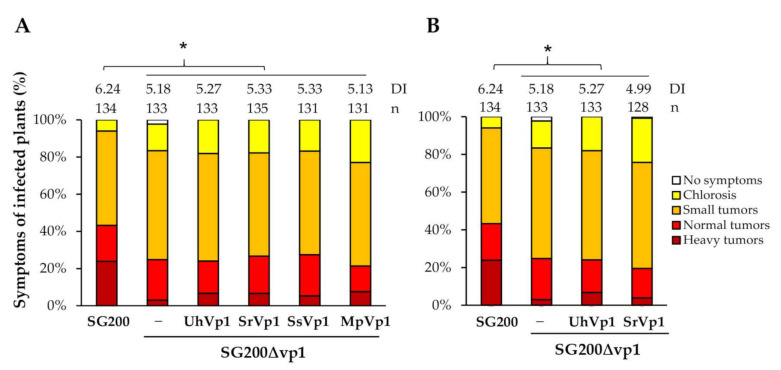
UmVp1 is functionally specialized in *U. maydis*. Pathogenicity assay of SG200Δvp1 complemented with respective orthologs from Ustilaginaceae, expressed under *umvp1* promoter using either their signal peptide (**A**) or UmVp1 signal peptide; (**B**) n indicates total numbers of infected plants from the three independent infection assays. DI is the average disease index value of three independent infection assays; * *p* < 0.05 indicates significant differences of disease symptoms in respective complementation strains compared with SG200 determined by a two-tailed Student’s *t*-test using the disease index values. Um, *Ustilago maydis* [16]; Uh, *Ustilago hordei* [37]; Sr, *Sporisorium reilianum* [7]; Ss, *Sporisorium scitamineum* [34].

## Data Availability

Gene expression data were retrieved from RNA-seq data [1] (GEO database: GSE103876). Genes and protein sequences are available at NCBI or MycoCosm (https://mycocosm.jgi.doe.gov/mycocosm/home).

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
