# Peer review of "A Novel Core Effector Vp1 Promotes Fungal Colonization and Virulence of Ustilago maydis"

_jof, 2021, doi:10.3390/jof7080589_

Round 1

Reviewer 1 Report

The authors have responded to all my comments and recommendations and performed several experiments to improve the Figure 2. With the current revised version, the submitted manuscript is suitable for publication.

Regards

Author Response

We appreciate your help in improving the manuscript.

Reviewer 2 Report

Comments from a reviewer

I would like to thank the authors for considering my comments from the last submission to improve the manuscript. The manuscript looks decent for me for publication. Please see below for a couple more of suggestions.

- Please consider an alternative word for different virulence assays. In my opinion, using the word ‘three biological replicates’ for the virulence assay may make the readers confused with ‘three replicates’ from other approaches like RT-PCR or Microscopy. I would prefer using the word ‘three independent assays’, but I will leave this up to the authors whether they want to change it.

- For statistical analysis of the virulence assay, please indicate which value the authors used for the student’s t-test. Did the authors use all values (% of each severity stage) for the test, or just select some of them to test? In this regard, I think the chi-square goodness of fit test would be more suitable to test the proportion of disease severity whether it is similar to a wild-type. This can be done by using the proportion of severity from the SG200 wild-type as an expected ratio, then testing across different mutant lines whether each observed proportion of severity fit to the ratio from the wild-type. My apology for suggesting an improper statistical approach in the last review.

Author Response

I would like to thank the authors for considering my comments from the last submission to improve the manuscript. The manuscript looks decent for me for publication. Please see below for a couple more of suggestions.

- Please consider an alternative word for different virulence assays. In my opinion, using the word ‘three biological replicates’ for the virulence assay may make the readers confused with ‘three replicates’ from other approaches like RT-PCR or Microscopy. I would prefer using the word ‘three independent assays’, but I will leave this up to the authors whether they want to change it.

We have now changed it to “three independent infection assays”

- For statistical analysis of the virulence assay, please indicate which value the authors used for the student’s t-test. Did the authors use all values (% of each severity stage) for the test, or just select some of them to test? In this regard, I think the chi-square goodness of fit test would be more suitable to test the proportion of disease severity whether it is similar to a wild-type. This can be done by using the proportion of severity from the SG200 wild-type as an expected ratio, then testing across different mutant lines whether each observed proportion of severity fit to the ratio from the wild-type. My apology for suggesting an improper statistical approach in the last review.

Thanks for pointing out. The disease index (DI) values from the three independent assays were used for the test. DI was the sum of values that calculated by multiplying the number of plants in a category with the value assigned to the category (death plant = 11, heavy tumor on base of stem or stunted growth = 9, tumor on leave and/or steam = 7, small tumors = 5, chlorosis = 3 and without symptoms = 1) and divided by the total infected plants per strain. We have now added in the section 2.1, and also indicated in the figure legends.   

We appreciate your comments and suggestions that helped improve the manuscript.

This manuscript is a resubmission of an earlier submission. The following is a list of the peer review reports and author responses from that submission.

Round 1

Reviewer 1 Report

The manuscript „The functional diverged effector Nle1 is required for Ustilago maydis proliferation and virulence” presents the identification of a secreted virulence factor of U. maydis.

Beside the virulence defect major findings are its experimental confirmation of its likely secretion and the identification of a nuclear localization signal at the c-terminus of the protein with a putative contribution to virulence. Furthermore the authors tested complementation of the virulence phenotype with a number of orthologs of Nle1 which all could not complement the phenotype.

The manuscript is well-structured and well written.   Most experiments were performed with proper controls, nevertheless, some of the data presented could also be interpreted differently and additional experiments will be required to justify the made interpretations and statements.

Major points:

  • The authors show that SG200 delta-nle1-gfp-umnle1 and also the c-terminal GFP-tagged Nle1 are both likely secreted. The signals visible are likely surrounding the hyphae in the biotrophic interphase (plasmolysis experiments would be nice to validate this). More importantly, I was surprised to see full complementation of the deletion strain with both GFP-tagged constructs.

If no nuclear GFP-signal in infected maize could be seen (I assume as otherwise such important data would have been shown), there are two options – either nle1 relevance in virulence plays in the apoplast and therefore the nls –part of the story is not really relevant or the Nle1 protein is processed in the apoplast and a fragment of it performs its virulence function within the maize cell…

Due to this important difference in interpreting the results I require a westernblot after maize infection to test if GFP gets cleaved off by some apoplastic protease or if the authors needs to consider an apoplastic function (and a name-change of the effector should be done consequently).

  • In Fig.3 the authors demonstrate that the mutation of the c-terminal NLS-motif leads to a reduced capacity to complement the phenotype. As a c-terminal tagging seems to be tolerated by the effector I would suggest 2 additional constructs to be tested for this part to make a clear result. Adding to the nls*mutant version of Nle1 either a plant NLS or a plant NES-motif + HA-tag, if the interpretation of the authors is correct the added NLS should lead to full rescue and the NES should lead to full inability to rescue the delta-nle1 phenotype.

Minor points:

  • The finding that none of the Nle1 homologs can rescue the virulence phenotype was indeed surprising. Testing at least the the Sporisorium SrNle1 strain for presence of protein expression would be a good thing to see if its an issue with the codon usage e.g.

The reasoning behind this point from my side is that your argumentation in the discussion that the functional divergence is a consequence of adapting to the respective host cell machinery, evading possibly also recognition etc… As Sporisorium reillianum has the same host as U. maydis the host-side cannot be a driving force (host cell machinery is the same to both fungi) (but of course it could be still the lifestyle). Therefore testing for protein-expression and levels in comparison to UmNle1 could ensure that we are not looking at a non-complementation due to an expression artifact.

  • Line 371 “over evolutionary… (I guess you want to write “time” but it was missed)

Author Response

The manuscript “The functional diverged effector Nle1 is required for Ustilago maydis proliferation and virulence” presents the identification of a secreted virulence factor of U. maydis.

Beside the virulence defect major findings are its experimental confirmation of its likely secretion and the identification of a nuclear localization signal at the c-terminus of the protein with a putative contribution to virulence. Furthermore, the authors tested complementation of the virulence phenotype with a number of orthologs of Nle1 which all could not complement the phenotype.

The manuscript is well-structured and well written.  Most experiments were performed with proper controls, nevertheless, some of the data presented could also be interpreted differently and additional experiments will be required to justify the made interpretations and statements.

We very much appreciate your positive comments, suggestions, and recognition of the significance of our work, and helps in improving the manuscript. We have changed the name of the effector to Vp1 (virulence promoting 1).

The authors show that SG200 delta-nle1-gfp-umnle1 and also the c-terminal GFP-tagged Nle1 are both likely secreted. The signals visible are likely surrounding the hyphae in the biotrophic interphase (plasmolysis experiments would be nice to validate this). More importantly, I was surprised to see full complementation of the deletion strain with both GFP-tagged constructs.

We failed to detect the GFP-fusion proteins in plasmolysis experiment. It could be caused by the unstable of GFP in low pH apoplasts. We performed plasmolysis experiments using the constitutively expressing strain SG200Δvp1-Potef-vp1-mCherry and used immunoblotting to detect the mCherry fusion proteins in total cell lysate and apoplastic fluid fractions of infected leaves. We could visualize the weaker fluorescence of Vp1-mCherry in the enlarged apoplastic space (Figure S4A).  And, we also detected the full-length Vp1-mCherry protein bands in the total cell lysates of infected leaves but the truncated bands in apoplastic fractions (Figure S4B). However, whether the truncated forms are the cause of unstable mCherry fusion proteins after secretion or the processing is not clear, because we also detected the truncated forms in the positive control- secreted mCherryHA proteins. Nonetheless, the results suggest the secretion of Vp1 in apoplasts. We have now included these results in the revised manuscript.

If no nuclear GFP-signal in infected maize could be seen (I assume as otherwise such important data would have been shown), there are two options – either nle1 relevance in virulence plays in the apoplast and therefore the nls –part of the story is not really relevant or the Nle1 protein is processed in the apoplast and a fragment of it performs its virulence function within the maize cell….

-We failed to visualize nuclear GFP-Vp1/Vp1-GFP fluorescence in infected leaves using the complementation strains expressing the fusion proteins under the native promoter or constitutive promoter otef (single copy and multiple copy integration). Based on our knowledge, so far, none of the translocated effectors of U. maydis (Tin2, Cmu1, Jsi1, See1 etc) could been successfully visualized inside the plant cells by microscopy. It is most likely due to the low amount of proteins translocated to plant cells.

Due to this important difference in interpreting the results I require a western blot after maize infection to test if GFP gets cleaved off by some apoplastic protease or if the authors need to consider an apoplastic function (and a name-change of the effector should be done consequently).

-Concerning the processing of Vp1 after secretion, we have now demonstrated that the C-terminus of Vp1 is processed. By adding HA tag at the C-terminus (Vp1-HA), we failed to visualize the protein bands in the supernatant fractions of cell culture and also the apoplastic fluid fractions (Figure 3D-E). However, when a larger tag was used (mCherry), we could not detect the full-length fusion proteins but truncated/processed fragments (Figure 3F). Unexpectedly, by deletion of NLS, we visualized the secretion of full-length Vp1(ΔNLS)-HA (Figure 3D-E). These results suggest a processing of Vp1 after secretion is occurred and is not depended on plant apoplastic protease. Although we could not demonstrate the translocation of Vp1 directly, we provide the indirect evidence to show the functionality of NLS in localizing Vp1 to the plant nucleus and the importance of NLS in Vp1 function (by deletion of nls). We agree that it is likely that at least a fragment of Vp1 might act inside the maize cells and/or also act in apoplast and might be not even translocated. However, if it is able to translocate, the functional NLS could guide the fragment into the plant nucleus. The results have now been included in the revised manuscript.

In Fig.3 the authors demonstrate that the mutation of the c-terminal NLS-motif leads to a reduced capacity to complement the phenotype. As a c-terminal tagging seems to be tolerated by the effector I would suggest 2 additional constructs to be tested for this part to make a clear result. Adding to the nls*mutant version of Nle1 either a plant NLS or a plant NES-motif + HA-tag, if the interpretation of the authors is correct the added NLS should lead to full rescue and the NES should lead to full inability to rescue the delta-nle1 phenotype.

We complemented Vp1(NLS*)-NLS-HA (NLS motif (PKKKRKV; derived from SV40 T antigen; Kalderon et al., 1984, Cell 39, 499-509) to Δvp1 mutant, and hoped that it could fully rescue the phenotype of Δvp1. Unfortunately, the Vp1(NLS*)-NLS-HA proteins also partially complement the reduced phenotype of Δvp1 to SG200 level (the attached figure) which is quite similar to the Vp1(NLS*) complementation strain. Considering the mutations on NLS does not affect the Vp1 processing, it is likely that Vp1(NLS*)-NLS-HA is still processing (Figure 3F) and other fragments could still be functional and rescue the virulence partially. By adding a copy of functional NLS after NLS*, it is not sure how it affects the localization of the proteins. As this data does not provide additional information, we decide against to include it in the revised manuscript but provide it here.

In the case of NLS deletion proteins Vp1(ΔNLS), Vp1(ΔNLS) could not complement the reduced phenotype of Δvp1 (Figure 3C) and the processing of Vp1(ΔNLS) is affected (Figure 3D&E).  Blocking the processing has significantly affected the virulence function of Vp1. We have now included these results and discussed in the revised manuscript.

The finding that none of the Nle1 homologs can rescue the virulence phenotype was indeed surprising. Testing at least the the Sporisorium SrNle1 strain for presence of protein expression would be a good thing to see if its an issue with the codon usage e.g. The reasoning behind this point from my side is that your argumentation in the discussion that the functional divergence is a consequence of adapting to the respective host cell machinery, evading possibly also recognition etc… As Sporisorium reillianum has the same host as U. maydis the host-side cannot be a driving force (host cell machinery is the same to both fungi) (but of course it could be still the lifestyle). Therefore testing for protein-expression and levels in comparison to UmNle1 could ensure that we are not looking at a non-complementation due to an expression artifact.

We totally agree with you. We were also surprised to find out none of the selected Vp1 orthologs could rescue the virulence phenotype. As we could not even detect the UmVp1 proteins which expressed under umvp1 native promoter (Δvp1-vp1HA or Δvp1-HAvp1; single or multiple copy integration), we could not check the protein levels of these orthologs we tested. As suggested, we performed qRT-PCR analysis, and found that the expression of srvp1 gene was up-regulated (the attached figure below). This result has now been added to the supplementary data (Figure S7).

Considering the protein levels of Vp1 orthologs could not be detected and the functions of orthologs have not been characterized, we thus tone down and suggest that U. maydis Vp1 acquired functionally specialized.

Line 371 “over evolutionary… (I guess you want to write “time” but it was missed)

We have now added it.

Reviewer 2 Report

Comments for authors

Major comments

In this article, the authors identified a novel virulent factor Nle1 in Ustilago maydis. The authors demonstrated that Nle1 is required for virulence in a maize host, Nle1 is localized in host nucleus, and its localization contributes to a partial virulence of the pathogen. Finally, the authors showed that Nle1 orthologs in other smut species do not contribute to virulence to the maize host.

I think the authors did a great job in terms of showing that Nle1 contributes to virulence and Nle1 is localized at the host nucleus. However, the authors did not provide further functional characterization to support what the authors discussed and concluded in this manuscript. The authors should either consider rewriting the discussion/conclusion or provide more evidence to support their statements. Here are a few examples to illustrate what I mean:

- Line 292ff: The authors stated that the nuclear localization signal of Nle1 is required for virulence function of U. maydis effector. However, the authors indicated that umnle1(nls*) shows an intermediate phenotype. The authors did not perform the statistical test to show ‘intermediate’ phenotype. The authors can show this in a couple of ways: do the t-test between umnle1(nls*) and nle1 mutant and umnle1(nls*) and SG200 wild-type or perform a post-hoc test for multiple treatments like the Tukey HSD test. If ‘intermediate’ phenotype is true, I would recommend rewrite to something like ‘the nls of Nle1 partially contributes to virulence function’.  To me, ‘required’ means that the function is completely abolished in the mutant. In addition, as Nle1 nuclear localization is not completely abolished in umnle1(nls*), the authors could delete the whole nls region to check the function.

- I am curious if Nle1 is directly required for fungal proliferation, or it just functions as an effector to weaken the host immunity and thus the pathogen has more proliferation and colonization. As Nle1 functions as a secreted protein, I think Nle1 has no direct effect on fungal proliferation, but instead has a function for fungal colonization. If the authors insist on their statement, the authors should provide more evidence regarding Nle1 function.

- Line 347ff: I think the reason why nle1 mutant does not affect the early stage of biotrophy is because Nle1 is a late-stage effector. This is shown in Lanver et al. 2018 data (Figure S1A) that Nle1 expression level is low before 4 dpi. As the authors did not perform a functional characterization of Nle1, I am skeptical when the authors say “the functional redundancy contributed by other effectors that target the different components of same Nle1-targeted pathway, which could compensate the absence of Nle1 at the earlier points in time.” How do the authors know, without functional mechanism of Nle1, that other effectors target different components in the same Nle1-targeted pathway?

- Did the authors check that Nle1 orthologs (Sr, Ss, Mp) are literally expressed in U. maydis recombinant strains? Maybe the reason why the orthologs do not contribute to the virulence as they are not expressed as proteins. Constructing transgenes from multiple species have a risk of getting non-functional or incomplete proteins unless the sequences are manually curated. The authors could check, at least by RT-qPCR, to verify that the orthologs are expressed.

- How do the authors assure true orthologs of Nle1 in other smut species? One-way BLAST search is a good method to identify gene homolog, but not true ortholog. The authors should provide more elaborated methods to prove true orthologs such as reciprocal best blast hit (BBH) method and/or gene tree-species tree reconciliation to make sure Nle1 homologs are diverged followed by the species tree phylogeny, which is a sign for ortholog. Other Nle1 homologs in U. maydis should be included in the analyses as the outgroup.

- The authors should have a separated section in Materials and Methods for “sequence analyses of UMAG00538”. In the section, the authors should mention how they perform protein domain analyses, signal peptide detection, NLS detection, Nle1 orthology finding and sequence alignment of orthologs from several smut fungi. Any results from signal/domain detection, like assigned function with scores and p-value, would be useful to verify their findings (these can be included in supplementary materials). Also, to credit persons who generate data for other smut fungi, please cite the references of all genomes/sequences the authors used for ortholog prediction of Nle1.

Minor comments

- Title: I am not sure if the ‘functional diverged effector’ is a good word as the authors did not provide an evidence about functional mechanism of Nle1.

- Keywords: The authors should use different words from the title to enhance visibility

- Line 29 – 31: I think this statement is true for biotrophic pathogens, but not all plant pathogenic fungi. Necrotrophic pathogens primarily secrete enzymes and/or toxins to destroy plant tissues, but not modulate plant immune response. Please consider rephrasing this. The authors can start a sentence as ‘Smut fungi’.

Line 72 – 74: Please indicate that this piece of information (an effector protein that is highly-upregulated during the biotrophic development) is retrieved from Lanver et al. 2018 study, and this is a rationale of the authors’ study to characterize Nle1.

Line 80, 195: The reference the authors cited is not the original study for SG200 strain. If wishing to cite the strain, please consider this publication:

Müller, P., Aichinger, C., Feldbrügge, M., & Kahmann, R. (1999). The MAP kinase kpp2 regulates mating and pathogenic development in Ustilago maydis. Molecular microbiology, 34(5), 1007-1017.

Line 81: Referencing tables should be in numerical order. Please consider changing to Table S1 and so on.

Line 81: Please consider changing the word to ‘potato dextrose agar (PDA)’.

-Section 2.2 for Plasmid and Strain construction: As there are many primers designed from this study, as well as several primer combinations. It would be great if the authors can provide a figure (maybe supplementary or overlayed with figure 1A) to show the gene region of Nle1 and locate where each primer binds to the gene region. This would be useful for readers to follow and in case to use them for further studies.

- Section 2.3, Table S1: Please indicate see1 primer sequences for RT-qPCR analyses

- Line 136: The tissues were ground in liquid nitrogen? Or how?

- Line 146 – 148: Were DNA from maize leaves normalized to the same concentration before RT-qPCR analyses?

- Line 153: Please spell out the full name for ‘TCA’.

- Line 193: Please consider rephrasing to ‘functionally characterized’.

- Figure S1B, Line 200 – 202: Please show the RT-qPCR data of nle1 and see1 from maize leaves to confirm organ-specificity of see1.

- Line 209: Please consider rephrasing to ‘either no tag or with HA-tag at the downstream of the N-terminus signal peptide’.

- Line 218: From represented photos, I am not sure if it is the leaf vein. Using ‘leaf tissue’ would be a better word choice if readers cannot judge that this is part of vascular tissues. Otherwise, please indicate a sign showing that it is the leaf vein.

- Figure 2A: Please indicate what ‘--' means. nle1 deletion mutant? Having bands for HANle1 without signal peptide would be very useful to confirm that Nle1 is a secreted protein.

- Figure 2C: I am not sure if the photos represent Nle1 secretion. Co-staining with dyes for fungal cell wall and plant plasma membrane would be useful to see the secretion. Also, is the Nle1-GFP or GFP-Nle1 signal observed in all hyphal tips? Or is it present at the biotrophic interface to plant plasma membrane?

- Line 247 – 253: The proteins were run in SDS-PAGE, which protein structure is supposed to be linearized. I am just curious why an anomalous migration can occur.

- Line 254: Please italicize ‘in planta’.

- Line 391ff: I am not convinced about functional diversification of Nle1 due to insufficient characterization data in other Nle1 orthologs.

Author Response

In this article, the authors identified a novel virulent factor Nle1 in Ustilago maydis. The authors demonstrated that Nle1 is required for virulence in a maize host, Nle1 is localized in host nucleus, and its localization contributes to a partial virulence of the pathogen. Finally, the authors showed that Nle1 orthologs in other smut species do not contribute to virulence to the maize host.

I think the authors did a great job in terms of showing that Nle1 contributes to virulence and Nle1 is localized at the host nucleus. However, the authors did not provide further functional characterization to support what the authors discussed and concluded in this manuscript. The authors should either consider rewriting the discussion/conclusion or provide more evidence to support their statements.

We thank the reviewer for the positive comments and the recognition of our work. After consider the translocation of the effector has not been demonstrated and it might act in apoplast, we now decide to change the name of the effector to Vp1 (virulence promoting 1).

Major comments

- Line 292ff: The authors stated that the nuclear localization signal of Nle1 is required for virulence function of U. maydis effector. However, the authors indicated that umnle1(nls*) shows an intermediate phenotype. The authors did not perform the statistical test to show ‘intermediate’ phenotype. The authors can show this in a couple of ways: do the t-test between umnle1(nls*) and nle1 mutant and umnle1(nls*) and SG200 wild-type or perform a post-hoc test for multiple treatments like the Tukey HSD test. If ‘intermediate’ phenotype is true, I would recommend rewrite to something like ‘the nls of Nle1 partially contributes to virulence function’.  To me, ‘required’ means that the function is completely abolished in the mutant. In addition, as Nle1 nuclear localization is not completely abolished in umnle1(nls*), the authors could delete the whole nls region to check the function.

Thanks for pointing it out and the suggestions. We have performed Student’s t-test between Δvp1-umvp1(nls*) and Δvp1 and umvp1(nls*) and SG200. As suggested, we have generated the complementation strain expressing nls deletion proteins (Vp1ΔNLS). The Vp1ΔNLS proteins could not be processed and the full-length proteins were detected in culture supernatant fraction and plant apoplasts, and it could not complement the phenotype of Δvp1 (Figure 3C-F). The results indicating the importance of NLS sequence in the Vp1 processing and the virulence function. We have included the results in the revised manuscript.

- I am curious if Nle1 is directly required for fungal proliferation, or it just functions as an effector to weaken the host immunity and thus the pathogen has more proliferation and colonization. As Nle1 functions as a secreted protein, I think Nle1 has no direct effect on fungal proliferation, but instead has a function for fungal colonization. If the authors insist on their statement, the authors should provide more evidence regarding Nle1 function.

We agree and have now rephrased the statement.

- Line 347ff: I think the reason why nle1 mutant does not affect the early stage of biotrophy is because Nle1 is a late-stage effector. This is shown in Lanver et al. 2018 data (Figure S1A) that Nle1 expression level is low before 4 dpi. As the authors did not perform a functional characterization of Nle1, I am skeptical when the authors say “the functional redundancy contributed by other effectors that target the different components of same Nle1-targeted pathway, which could compensate the absence of Nle1 at the earlier points in time.” How do the authors know, without functional mechanism of Nle1, that other effectors target different components in the same Nle1-targeted pathway?

Thank you for pointing out this. We have removed the part and rewrite it.

- Did the authors check that Nle1 orthologs (Sr, Ss, Mp) are literally expressed in U. maydis recombinant strains? Maybe the reason why the orthologs do not contribute to the virulence as they are not expressed as proteins. Constructing transgenes from multiple species have a risk of getting non-functional or incomplete proteins unless the sequences are manually curated. The authors could check, at least by RT-qPCR, to verify that the orthologs are expressed.

We acknowledge your concerns and thank for your suggestion. We have now performed qRT-PCR analysis to check on srvp1 ortholog gene expression and found that the expression of srvp1 gene was up-regulated (the attached figure below). This shows that at least SrVp1 has no problem in expressing in U. maydis recombinant strain. This result has been added to the supplementary data (Figure S7). Considering the protein levels of Vp1 orthologs could not be detected, we thus tone down and suggest that Vp1 orthologs might be functional diverse.

- How do the authors assure true orthologs of Nle1 in other smut species? One-way BLAST search is a good method to identify gene homolog, but not true ortholog. The authors should provide more elaborated methods to prove true orthologs such as reciprocal best blast hit (BBH) method and/or gene tree-species tree reconciliation to make sure Nle1 homologs are diverged followed by the species tree phylogeny, which is a sign for ortholog. Other Nle1 homologs in U. maydis should be included in the analyses as the outgroup.

Thanks for the suggestion. We have now performed reciprocal best blast hit (BBH) and also retrieved the Vp1 ortholog information from the ortholog database search OrthoDB. We have added this information in the results and described in “Materials and Methods” sections of the revised manuscript.

- The authors should have a separated section in Materials and Methods for “sequence analyses of UMAG00538”. In the section, the authors should mention how they perform protein domain analyses, signal peptide detection, NLS detection, Nle1 orthology finding and sequence alignment of orthologs from several smut fungi. Any results from signal/domain detection, like assigned function with scores and p-value, would be useful to verify their findings (these can be included in supplementary materials). Also, to credit persons who generate data for other smut fungi, please cite the references of all genomes/sequences the authors used for ortholog prediction of Nle1.

We have now added the section 2.7- Bioinformatic Analysis. The score values given by SignalP, the predicted NLS sequences by LOCALIZER, the coverage percentage and identity percentage, and corresponding E-values are provided in the supplementary data (Figure S3). The references have been cited.

Minor comments

- Title: I am not sure if the ‘functional diverged effector’ is a good word as the authors did not provide an evidence about functional mechanism of Nle1.

We have removed it and changed the title.  

- Keywords: The authors should use different words from the title to enhance visibility

We have modified it.

- Line 29 – 31: I think this statement is true for biotrophic pathogens, but not all plant pathogenic fungi. Necrotrophic pathogens primarily secrete enzymes and/or toxins to destroy plant tissues, but not modulate plant immune response. Please consider rephrasing this. The authors can start a sentence as ‘Smut fungi’.

We have changed it.

- Line 72 – 74: Please indicate that this piece of information (an effector protein that is highly-upregulated during the biotrophic development) is retrieved from Lanver et al. 2018 study, and this is a rationale of the authors’ study to characterize Nle1.

We have mentioned and cited it in the first three lines of the result section 3.1 and in the supplementary data (Figure S1A).

- Line 80, 195: The reference the authors cited is not the original study for SG200 strain. If wishing to cite the strain, please consider this publication:

Müller, P., Aichinger, C., Feldbrügge, M., & Kahmann, R. (1999). The MAP kinase kpp2 regulates mating and pathogenic development in Ustilago maydis. Molecular microbiology, 34(5), 1007-1017.

We have cited the reference.

- Line 81: Referencing tables should be in numerical order. Please consider changing to Table S1 and so on.

It has been changed.

- Line 81: Please consider changing the word to ‘potato dextrose agar (PDA)’.

We have changed it.

-Section 2.2 for Plasmid and Strain construction: As there are many primers designed from this study, as well as several primer combinations. It would be great if the authors can provide a figure (maybe supplementary or overlayed with figure 1A) to show the gene region of Nle1 and locate where each primer binds to the gene region. This would be useful for readers to follow and in case to use them for further studies.

We have tried but found it more complicated, and decided to keep as in Table S1 and S2.

- Section 2.3, Table S1: Please indicate see1 primer sequences for RT-qPCR analyses

We have added them.

- Line 136: The tissues were ground in liquid nitrogen? Or how?

We have now added it.

- Line 146 – 148: Were DNA from maize leaves normalized to the same concentration before RT-qPCR analyses?

The sentence “All of the cDNA samples were diluted to 20ng/ul before performing qRT-qPCR analysis” has been added.

- Line 153: Please spell out the full name for ‘TCA’.

It has been changed.

- Line 193: Please consider rephrasing to ‘functionally characterized’.

It has been rephrased.

- Figure S1B, Line 200 – 202: Please show the RT-qPCR data of nle1 and see1 from maize leaves to confirm organ-specificity of see1.

We have now included the see1 gene expression in maize leaves retrieved from RNAseq dataset (Lanver et. al, 2018) (Figure S1A), and the see1 gene expression in maize tassels (Figure S1B).

- Line 209: Please consider rephrasing to ‘either no tag or with HA-tag at the downstream of the N-terminus signal peptide’.

It has been rephrased.

- Line 218: From represented photos, I am not sure if it is the leaf vein. Using ‘leaf tissue’ would be a better word choice if readers cannot judge that this is part of vascular tissues. Otherwise, please indicate a sign showing that it is the leaf vein.

The text has been modified as suggested.

- Figure 2A: Please indicate what ‘--' means. nle1 deletion mutant? Having bands for HANle1 without signal peptide would be very useful to confirm that Nle1 is a secreted protein.

Thanks for your suggestion. We have generated the strain SG200Δvp1 constitutively expressing HAVp1 without signal peptide (dSP-HAVp1) as a negative control for secretion. The proteins were expressed and detected in the cell pellet fractions but not in supernatant (Figure 2A). We have included in the revised manuscript.

We have indicated ‘-‘ as SG200 strain in the figure legend.

- Figure 2C: I am not sure if the photos represent Nle1 secretion. Co-staining with dyes for fungal cell wall and plant plasma membrane would be useful to see the secretion. Also, is the Nle1-GFP or GFP-Nle1 signal observed in all hyphal tips? Or is it present at the biotrophic interface to plant plasma membrane?

We have now performed the plasmolysis experiments to expand the apoplastic space and also the immunoblotting analysis of apoplastic fluid fractions collected from infected leaves to support the secretion of the Vp1 proteins. These results have now been included in the supplementary figure S4 (see response to the reviewer#1).

- Line 247 – 253: The proteins were run in SDS-PAGE, which protein structure is supposed to be linearized. I am just curious why an anomalous migration can occur.

Disordered proteins are often found to migrate slower than expected. We’re also not sure how it could happen and could not provide further information. We have provided our speculation about the anomalous migration in the revised manuscript.

- Line 254: Please italicize ‘in planta’.

We have changed it.

- Line 391ff: I am not convinced about functional diversification of Nle1 due to insufficient characterization data in other Nle1 orthologs.

We have rewritten the manuscript and modified the text to avoid overstatements. Thanks for your comments and suggestions.

Reviewer 3 Report

Review points

Title: The functional diverged effector Nle1 is required for Ustilago maydis proliferation and virulence

In their paper, Cuong Van Hoan and Lay-Sun Ma showed that Ustilago maydis secretes Nle1 effector that contributes to the full virulence of this smut fungus during maize colonization. In this regard, the authors knocked-out Nle1 in U. maydis SG200 background and performed disease assay on the host plant, which showed reduced virulence phenotype. They also showed that SG200 expressing SP-Nle1-GFP recombinant protein shows fluorescent signal around the colonized hypha, indicating secretion of this effector. Moreover, they predicted C-terminal NLS signal that is important for virulence function of Nle1 protein and for its localization in the host cell nucleus.

Last but not least, they found orthologs of Nle1 protein, which shows diversification in C-terminal and NLS regions, in other smut fungi. Complementation of orthologous Nle1 genes from other smut fungi in SG200∆nle1 mutant background did not rescue the reduced virulence phenotype, indicating functional diversification of Nle1 protein n different smut fungi.

Minor points:

Line 12, 18, 31, 354, 356, 392: please use   ‘, which’ instead of ‘which’  (use comma before which)

Line 14: please use ‘fungus’ instead of ‘fungi’

Line 32: instead of ‘genes are encoded for the signal peptide containing secreted proteins’, please use ‘genes encode proteins containing a signal peptide for secretion’.

Line 34: instead of ‘and secrete via an unconventional secretion pathway’, please use ‘and are predicted to be secreted via an unconventional secretion pathway’.

Line 35: instead of ‘It makes’ please use ‘This makes’

Line 13, 35: instead of U. maydis’ please useUstilago maydis’

Line 40, 42:    , while  (use comma before while)

Line 51: instead of The corn smut Ustilago maydis induce anthocyanin and form tumorsplease useThe corn smut U. maydis induces anthocyanin and forms tumors

Line 53: instead of ‘fungal development’ please use ‘fungal pathogenic development’

Line 157: Please indicate whether any SP was used for subcellular localization assay of Nle1-GFP

Line 183: instead of ‘including Umag00538’ please use ‘,including umag00538,’

Line 185: instead of ‘virulent factor’ please use ‘virulence factor’

Line 193: instead of ‘we functional characterize’ please use ‘we functionally characterized’

Line 203: instead of ‘the expression of nle1 gene is biotrophic induced and’ please use ‘the expression of nle1 gene is induced during biotrophic phase and’

Line 204: instead of ‘contribution of Nle1 on U. maydis virulence’ please use ‘contribution of Nle1 to U. maydis virulence’

Line 212: instead of ‘in plant cells’ please use ‘in plant leaves’

Line 217: please integrate the method in the text ‘via WGA-AF488/PI staining’

Line 219: instead of ‘4dpi’ please use ‘4 dpi’

Line 220: instead of ‘The consistent findings from the fungal biomass analysis and the collecting microscopic data demonstrate that’ please use ‘both fungal biomass analysis and microscopic data analysis demonstrate that’

Line 275: Please indicates in the text whether any SP was used for localization of Nle1-GFP

Line 306: Please refer ‘Figure S3’ in the text to indicate which Nle1 orthologs has NLS

Line 392: Please delete ‘general’

Line 224: For Figure1 A, there is no predicted domain for Nle1; therefore, I think to say ‘Domain organization’ would be wrong.

Major points:

  1. For in vitro and in planta Nle1 localization assay (Figure 2A and C) the proper negative and positive controls are missing. For figure 2A, authors should use HANle1 without signal peptide for negative control instead of wt SG200 strain. And for Figure 2C, internal GFP expressing SG200 should be used as a negative control.

Author Response

1
Title: The functional diverged effector Nle1 is required for Ustilago maydis proliferation and virulence
In their paper, Cuong Van Hoang and Lay-Sun Ma showed that Ustilago maydis secretes Nle1 effector that contributes to the full virulence of this smut fungus during maize colonization. In this regard, the authors knocked-out Nle1 in U. maydis SG200 background and performed disease assay on the host plant, which showed reduced virulence phenotype. They also showed that SG200 expressing SP-Nle1-GFP recombinant protein shows fluorescent signal around the colonized hypha, indicating secretion of this effector. Moreover, they predicted C-terminal NLS signal that is important for virulence function of Nle1 protein and for its localization in the host cell nucleus.
Last but not least, they found orthologs of Nle1 protein, which shows diversification in C-terminal and NLS regions, in other smut fungi. Complementation of orthologous Nle1 genes from other smut fungi in SG200Δnle1 mutant background did not rescue the reduced virulence phenotype, indicating functional diversification of Nle1 protein n different smut fungi.
Minor points:
Line 12, 18, 31, 354, 356, 392: please use ‘, which’ instead of ‘which’ (use comma before which)
We have changed them.
Line 14: please use ‘fungus’ instead of ‘fungi’
It has been changed.
Line 32: instead of ‘genes are encoded for the signal peptide containing secreted proteins’, please use ‘genes encode proteins containing a signal peptide for secretion’.
It has been changed.
Line 34: instead of ‘and secrete via an unconventional secretion pathway’, please use ‘and are predicted to be secreted via an unconventional secretion pathway’.
We have changed it.
Line 35: instead of ‘It makes’ please use ‘This makes’
We have changed it.
Line 13, 35: instead of ‘U. maydis’ please use ‘Ustilago maydis’
We have changed it.
Line 40, 42: , while (use comma before while)
It has been changed.
Line 51: instead of ‘The corn smut Ustilago maydis induce anthocyanin and form tumors’ please use ‘The corn smut U. maydis induces anthocyanin and forms tumors’
2
It has been changed.
Line 53: instead of ‘fungal development’ please use ‘fungal pathogenic development’
It has been changed.
Line 157: Please indicate whether any SP was used for subcellular localization assay of Nle1-GFP
It has been indicated in the figure legend and the text.
Line 183: instead of ‘including Umag00538’ please use ‘,including umag00538,’
It has been changed.
Line 185: instead of ‘virulent factor’ please use ‘virulence factor’
It has been changed.
Line 193: instead of ‘we functional characterize’ please use ‘we functionally characterized’
It has been changed.
Line 203: instead of ‘the expression of nle1 gene is biotrophic induced and’ please use ‘the expression of nle1 gene is induced during biotrophic phase and’
It has been changed.
Line 204: instead of ‘contribution of Nle1 on U. maydis virulence’ please use ‘contribution of Nle1 to U. maydis virulence’
It has been changed.
Line 212: instead of ‘in plant cells’ please use ‘in plant leaves’
It has been changed.
Line 217: please integrate the method in the text ‘via WGA-AF488/PI staining’
It has been changed.
Line 219: instead of ‘4dpi’ please use ‘4 dpi’
It has been changed.
Line 220: instead of ‘The consistent findings from the fungal biomass analysis and the collecting microscopic data demonstrate that’ please use ‘both fungal biomass analysis and microscopic data analysis demonstrate that’
We have rephrased it.
Line 275: Please indicates in the text whether any SP was used for localization of Nle1-GFP
It has been changed.
Line 306: Please refer ‘Figure S3’ in the text to indicate which Nle1 orthologs has NLS
We have added the information in Figure S3C.
Line 392: Please delete ‘general’
3
It has been removed.
Line 224: For Figure1 A, there is no predicted domain for Nle1; therefore, I think to say ‘Domain organization’ would be wrong.
We have changed it to ‘Schematic drawing of Vp1’
Major points:
For in vitro and in planta Nle1 localization assay (Figure 2A and C) the proper negative and positive controls are missing. For figure 2A, authors should use HANle1 without signal peptide for negative control instead of wt SG200 strain. And for Figure 2C, internal GFP expressing SG200 should be used as a negative control."
These two controls-dSP-HAVp1 and SG200-GFP (no SP) are now included in the Figure 2A and 2C, respectively. Thanks for your suggestions.